# The effect of metabolic health and obesity on lung function: A cross sectional study of 114,143 participants from Kangbuk Samsung Health Study

**Jonghoo Lee[1]**, **Hye Kyeong Park[2]**, **Min-Jung Kwon[3]**, **Soo-Youn Ham[4]**, **Si-Young Lim[5]**, **Jae-Uk Song[5]***

**1** Department of Internal Medicine, Jeju National University Hospital, Jeju National University School of Medicine, Jeju, Republic of Korea, **2** Division of Pulmonary and Critical Care Medicine, Department of Internal Medicine, Ilsan Paik Hospital, Inje University College of Medicine, Ilsan, Republic of Korea, **3** Department of Laboratory Medicine, Kangbuk Samsung Hospital, Sungkyunkwan University School of Medicine, Seoul, Republic of Korea, **4** Department of Radiology, Kangbuk Samsung Hospital, Sungkyunkwan University School of Medicine, Seoul, Republic of Korea, **5** Division of Pulmonary and Critical Care Medicine, Department of Internal Medicine, Kangbuk Samsung Hospital, Sungkyunkwan University School of Medicine, Seoul, Republic of Korea

☉ These authors contributed equally to this work.

\* khfever76@gmail.com

## Abstract

### Objective

Although the role of obesity-induced metabolic abnormalities in impaired lung function is well-established, the risk of impaired lung function among obese individuals without metabolic abnormalities, referred to metabolically-healthy obesity (MHO), is largely unexplored. Therefore, we evaluated the impact of MHO on lung function in a large health-screening cohort.

### Methods

114,143 subjects (65,342 men, mean age and BMI: 39.6 years and 23.6) with health examinations in 2019 were divided into four groups as follows: metabolically healthy non-obese (MHNO), MHO, metabolically unhealthy non-obese (MUHNO), and metabolically unhealthy obese (MUHO). Metabolic health was defined as fewer than two metabolic syndrome components. Obesity was defined as BMI $\geq$25 kg/m$^2$. Adjusted odds ratios (aORs), using MHNO as a reference, were calculated to determine lung function impairment.

### Results

Approximately one-third (30.6%) of the study subjects were obese. The prevalence of MHO was 15.1%. Subjects with MHO had the highest FEV1% and FVC% values but the lowest FEV1/FVC ratio ($p$<0.001). These results persisted after controlling for covariates. Compared with MHNO, the aORs (95% confidence interval) for FEV1% < 80% in MHO, MUHNO and MUHO were 0.871 (0.775–0.978), 1.274 (1.114–1.456), and 1.176 (1.102–1.366),

**Data Availability Statement:** All relevant data are within the paper. And, due to ethical restrictions imposed by the Institutional Review Board of

Kangbuk Samsung Hospital, the patient data are not available for distribution outside of the Kangbuk Samsung Hospital. For additional information, please contact Gayoung Lim (gayoung.lim@samsung.com) or the Institutional Review Board of Kangbuk Samsung Hospital: Address: 29 Saemunan-ro, Jongno-Gu, Seoul, Korea (03181) E-mail: irb.kbsmc@samsung.com Telephone: 82-2-2001-1943, 1945 - Fax: 82-2-2001-1946.

**Funding:** One of authors (Jonghoo Lee) was supported by 2022 scientific promotion program funded by Jeju National University. However, the funders had no role in study design, data collection and analysis, decision to publish, or preparation of the manuscript.

**Competing interests:** The authors declare that they have no conflict of interest.

respectively (P for trend = 0.014). Similarly, the aORs in MHO, MUHNO, and MUHO were 0.704 (0.615–0.805), 1.241 (1.075–1.432), and 1.226 (1.043–1.441), respectively, for FVC % < 80% (p for trend = 0.013). However, the aORs for FEV1/FVC<0.7 were not significantly different between groups (*p* for trend = 0.173).

## Conclusions

The MHO group had better lung function than other groups. However, longitudinal follow-up studies are required to validate our findings.

## Introduction

The worldwide prevalence of obesity has increased dramatically, making obesity a major public health concern because of the documented risks of cardio-metabolic diseases and mortality [1, 2]. Despite its typical health risks, obesity is also counterintuitively associated with protection against cardio-metabolic diseases [2]. There is increasing recognition that obese individuals can have favorable metabolic profiles, a condition referred to as metabolically healthy obesity (MHO), which seems to be harmless. However, debate remains regarding whether MHO is truly healthy [1, 3].

Obesity also impacts lung function [4]. However, previous studies have produced mixed findings regarding the impact of obesity on lung function, with some studies finding a negative relationship [5–7] but others not [8, 9], although the role of obesity-induced metabolic abnormalities in impaired lung function is well-established [10]. Furthermore, it is unclear how much obesity related lung effects occur independent of metabolic abnormalities [11].

Only one study to date has attempted to evaluate the relationship between MHO and lung function [12]. However, this study used Choi's reference equations, which have not been validated for normal populations and are known to present differences from observed values [13]. Furthermore, about one third of the study subjects had diabetes (34%) and hypertension (37.7%) both of which are themselves related to impaired lung function [14], regardless of metabolic unhealthy (MUH). Consequently, the previous study alone is not sufficient to determine the effect of MHO on lung function. Therefore, we compared spirometric values between subjects of different metabolic health and obesity status to define the impact of MHO on lung function in a large asymptomatic population.

## Materials and methods

### Study design and population

This cross-sectional study was a part of the Kangbuk Samsung Health Study (KSHS), in which subjects participated in a medical health checkup program at the Health Promotion Center of Kangbuk Samsung Hospital. The comprehensive health-screening program assessed demographic, anthropometric, and laboratory data. In Republic of Korea, the Industrial Safety and Health Law requires all employees to participate in annual or biennial health examinations that are offered free of charge. Most of the examinees were employees or family members of companies or local governmental organizations. The remaining participants voluntarily registered for screening examinations.

We initially included 214,551 individuals who underwent comprehensive health examinations in 2019. From these, we extracted 212,333 participants with recorded spirometry and

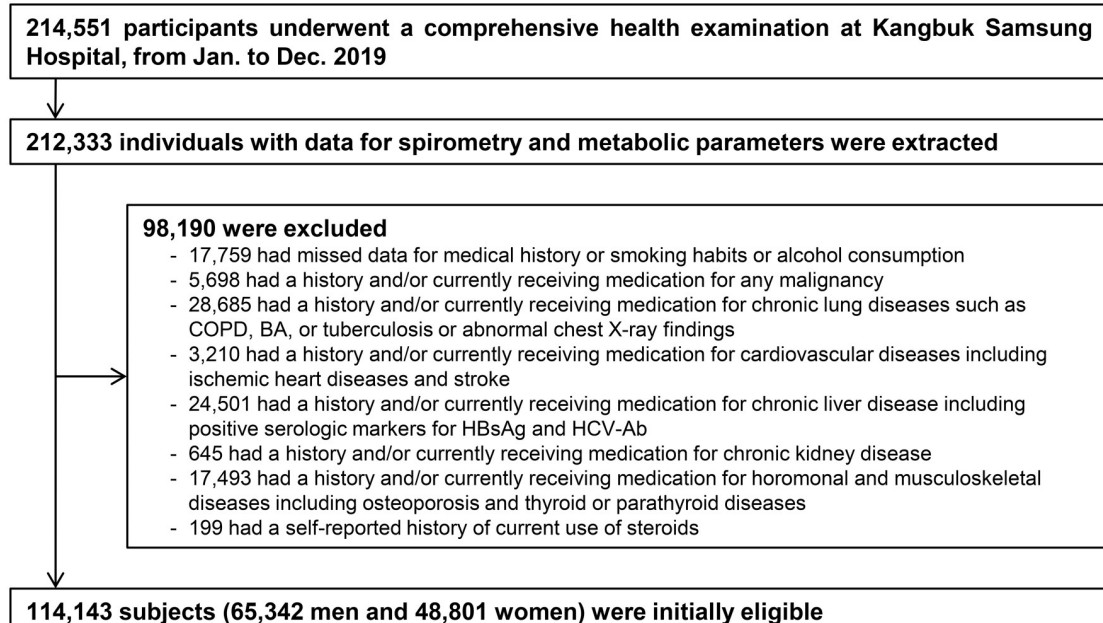

**Fig 1. Selection of study participants.** BA = bronchial asthma; COPD = chronic obstructive pulmonary disease; HBsAg = hepatitis B virus surface antigen; HCV-Ab = hepatitis C virus antibody.

metabolic data used to ascertain metabolic health. Among the potential subjects, we excluded those with missing data for medical history and smoking habits or alcohol consumption (n = 17,759). We additionally excluded 80,431 participants with either self-reported histories and/or who were currently receiving medications for any medical conditions. Detailed comorbidities were unavailable, because the medical history questionnaire only required yes/no responses. As some individuals had more than one exclusion criterion, 114,143 subjects were eligible for final analysis (Fig 1).

The study was approved by the Institutional Review Board of Kangbuk Samsung Hospital and was in accordance with the Helsinki Declaration of 1975. The requirement for informed consent was waived, as we retrospectively accessed a de-identified database for analysis purposes.

### Exposures: Metabolic health and obesity status

Physical characteristics and serum biochemical parameters were measured by trained nurses. Height and weight were calculated with individuals wearing light clothing and bare feet, using automated instruments (InBody 3.0 and Inbody 720, Biospace Co., Seoul, Korea) that were validated for reproducibility and accuracy of body composition measurements [15] and were calibrated every morning before testing. Body mass index (BMI) was calculated as weight in kg divided by square of height in m. Blood pressure (BP) was measured with a standard sphygmomanometer after at least 5 min of seated rest. Measurements were performed twice at 5 min intervals and averaged for analysis.

Blood samples were collected after a 10 h fast. Serum total cholesterol and triglycerides were determined with an enzymatic colorimetric assay. Low-density lipoprotein cholesterol (LDL-C) and high-density lipoprotein cholesterol (HDL-C) were measured directly with a homogeneous enzymatic colorimetric assay. Serum glucose was measured using the

hexokinase method on a Cobas Integra 800 apparatus (Roche Diagnostics). The inter- and intraassay coefficients of variation for quality control specimens were <5% for all blood variables. Insulin resistance was assessed using the homeostasis model assessment of insulin resistance (HOMA-IR) equation: fasting blood insulin (μU/ml) × fasting blood glucose (mmol/l)/ 22.5 [16]. The Laboratory Medicine Department at Kangbuk Samsung Hospital has been accredited and participates annually in inspections and surveys by the Korean Association of Quality Assurance for Clinical Laboratories. Obesity was defined as BMI ≥25 kg/m$^2$ (the proposed cut-off for Asian populations) [1, 17–19]. Metabolic health (MH) was defined as having fewer than two of the following risk factors [20]: elevated blood pressure (≥ 130/85) or antihypertensive drugs, high triglycerides (≥150 mg/dl) or lipid-lowering drugs, high fasting glucose (≥100 mg/dl) or medications for diabetes, low HDL-cholesterol (<40 mg/dL in men, <50 mg/dL in women), and HOMA-IR ≥90th percentile (≥2.9). MUH was defined as having two or more of the metabolic abnormalities described above. We used this definition alongside obesity status to create four phenotypes: metabolically healthy, non-obese (MHNO), metabolically healthy, obese (MHO), metabolically unhealthy, non-obese (MUHNO) and metabolically unhealthy, obese (MUHO).

## Covariates

Our analyses were adjusted for socio-demographic variables, behavioral factors and laboratory parameters. Information on demographic characteristics, smoking habits, alcohol intake (g/ day), exercise frequency, medical history, medication use, and education level were acquired using standardized, self-administered questionnaires. Smoking status was classified as non-smoker, ex-smokers (any prior regular use), and current smoker (current use). Alcohol consumption was categorized as non-heavy (≤20g/day) and heavy (>20g/day). Weekly frequency of moderate physical activity (defined as more than 30 min of activity inducing slight breathlessness per day) was also assessed, and regular exercise was defined as ≥3 times/week [17]. Education level was categorized as less than college graduate or college graduate or more [17]. Additionally, serum levels of liver enzymes, creatinine, and high-sensitivity C-reactive protein (hsCRP) were measured as described previously [18, 19].

## Outcomes: Lung function

Spirometry was performed as recommended by the American Thoracic Society [21] using the Vmax22 system (Sensor-Medics, Yorba Linda, CA). FEV1 and FVC were obtained under a pre-bronchodilatory setting. The highest forced expiratory volume in 1s (FEV1) and forced vital capacity (FVC) values from three or more tests with acceptable curves were used for further analyses. The predicted values for FEV1 and FVC were calculated using equations for a representative Korean population sample [22]. To calculate the predicted FVC% (FVC%) and predicted FEV1% (FEV1%), we divided the measured value (L) by the predicted value (L) and converted the quotient into a percentage. The ratio of FEV1 to FVC (FEV1/FVC) was also calculated using the actual measurement. The following criteria were used to determine impaired lung function: FEV1%<80%, FVC%<80%, and FEV1/FVC<0.7 (refers to obstructive lung function, OLF) [23].

## Statistical analyses

Data are presented as mean ± standard deviations or median and interquartile range for continuous variables and as proportions for categorical variables. The distribution of continuous variables was assessed with the Kolmogorov-Smirnov test. The baseline continuous variables were stratified by metabolic health and obesity status and compared using one-

way analysis of variance (ANOVA) or Kruskal-Wallis tests. The chi-square test or Fisher's exact test was used for categorical variables. Analysis of covariance (ANCOVA) was performed to test differences of mean values of lung function parameters between study groups after adjusting for age, sex, and continuous variables with $p < 0.05$ in univariate analyses. Post-hoc analysis was performed using the Bonferroni correction to compare the mean spirometric values between study groups. To analyze the significance of differences among groups according to metabolic health and obesity status, all covariates were treated as categorical variables: high or low and with or without. Differences among groups were tested using chi square or Fisher's exact test.

Binary logistic regression was used to assess the effects of metabolic health and obesity on lung function impairment. We estimated the adjusted odds ratios (aORs) with 95% confidence intervals (CI) for impairment of lung function in MHO, MUHNO, and MUHO compared with MHNO as the reference group. We fitted three models with progressive adjustments for potential confounding factors; model 1 was adjusted for age, sex, smoking status, alcohol intake, regular exercise, and education level; model 2 was adjusted as in model 1 plus glucose, total cholesterol, triglycerides, HDL-C, LDL-C, HOMA-IR, and systolic BP; model 3 was adjusted as in model 2 plus variables with $p < 0.05$ in univariate analyses. As FVC (L) and FEV1 (L) were strongly correlated (r = 0.942, $p < 0.001$), these parameters were assessed separately to avoid confounding effects. All tests were two-sided, and $p$ values$< 0.05$ were considered statistically significant. Data were analyzed using IBM SPSS Statistics 19.0 (IBM, Armonk, NY, USA).

## Results

### Baseline characteristics of participants

The characteristics of the 114,143 eligible subjects (57.2% male, 39.6±7.8 years) are summarized in Table 1. Mean BMI was 23.6±3.3. Approximately one third (n = 34,877, 30.6%) of study subjects were obese, but all were moderately obese. Classification of subjects according to metabolic health and obesity status showed that 58.9% of the subjects were MHNO, 15.1% were MHO, 10.5% were MUHNO, and 15.5% were MUHO. MUH groups showed significantly worse total cholesterol, triglycerides, HDL-C, LDL-C, fasting glucose, HOMA-IR, and BP than did the MH groups. Regarding spirometric values (Table 2), FEV1, FEV1 (%), FVC, and FVC (%) were higher in the MH groups than MUH groups after adjusting for multiple covariates. However, FEV1/FVC was lower in the obese group than in the non-obese group. In particular, the subjects with MHO had the highest spirometric values but the lowest FEV1/FVC among the four groups ($p < 0.001$).

### Clinical and laboratory parameters according to metabolic health and obesity status

A comparison of clinical and laboratory characteristics among groups divided by metabolic health and obesity status is shown in Table 3. MUH groups were older and more likely to smoke and drink alcohol. Regarding MUH parameters, the prevalence of all components was higher in MUH groups compared with MH groups. In contrast, more subjects exercised regularly in MH groups compared with MUH groups. Especially, the MHO group included the highest proportion of subjects who exercised regularly. The MH groups had more subjects with higher education. Proportions of subjects with FEV1%<80% and FVC%<80% were significantly higher in the MUH groups compared with MH counterparts. However, obese groups had significantly more frequent OLF than non-obese groups.

**Table 1. Baseline characteristics of study participants classified by metabolic health and obesity status.**

| | All Subjects (N = 114,143) | MHNO (N = 67,261, 58.9%) | MHO (N = 17,192, 15.1%) | MUHNO (N = 12,005, 10.5%) | MUHO (N = 17,685, 15.5%) | *p* value |
|---|---|---|---|---|---|---|
| **Age (years)** | 39.6±7.8 | 38.5±7.5 | 39.2±7.4 | 43.3±8.5 | 41.5±7.9 | <0.001 |
| **Sex (male)** | 65,342 (57.2) | 28,653 (42.6) | 13,226 (76.9) | 8,671 (72.2) | 14,792 (83.6) | <0.001 |
| **Height (cm)** | 168.2±8.4 | 166.5±8.2 | 170.6±8.1 | 169.4±8.2 | 171.4±7.7 | <0.001 |
| **Weight (kg)** | 67.1±13.1 | 60.3±9.4 | 78.5±9.0 | 66.4±8.4 | 82.5±10.6 | <0.001 |
| **BMI (kg/m$^2$)** | 23.6±3.3 | 21.6±2.0 | 26.9±1.8 | 23.1±1.5 | 28.0±2.6 | <0.001 |
| **Smoking status** | | | | | | <0.001 |
| Non smoker | 6,483 (57.4) | 45,789 (68.1) | 8,236 (47.9) | 5,219 (43.5) | 6,239 (35.3) | |
| Former smoker | 29,669 (26.0) | 13,509 (20.1) | 5,474 (31.8) | 4,113 (34.3) | 6,573 (37.2) | |
| Current smoker | 18,991 (16.6) | 7,963 (11.8) | 3,482 (20.3) | 2,673 (22.3) | 4,873 (27.6) | |
| **Smoking (pack-years)** | 3.8±7.3 | 2.4±5.7 | 4.6±7.3 | 6.2±9.4 | 7.1±9.4 | <0.001 |
| **Amount of alcohol consumption (g/day)** | 13.9±21.5 | 10.9±17.8 | 16.5±23.4 | 17.0±25.3 | 20.4±26.6 | <0.001 |
| **Moderate physical activity frequency (times/week)** | 0.87±1.42 | 0.83±1.41 | 1.05±1.50 | 0.81±1.39 | 0.85±1.37 | <0.001 |
| **High education (≥college graduate) (n = 111,747)** | 94,181 (84.3) | 56,457 (85.5) | 14,339 (85.3) | 9,394 (80.2) | 13,991 (81.2) | <0.001 |
| **Total bilirubin (mg/dL)** | 0.80±0.36 | 0.79±0.36 | 0.83±0.36 | 0.79±0.36 | 0.79±0.35 | <0.001 |
| **ALT (U/L) (n = 113,969)** | 18.0 (13.0–27.0) | 15.0 (11.0–21.0) | 22.0 (16.0–32.0) | 21.0 (15.0–30.0) | 30.0 (21.0–45.0) | <0.001 |
| **Serum creatinine (mg/dL)** | 0.81±0.17 | 0.77±0.17 | 0.87±0.16 | 0.85±0.18 | 0.88±0.16 | <0.001 |
| **Total cholesterol (mg/dL)** | 195.4±33.8 | 190.2±31.2 | 199.4±32.1 | 202.8±37.7 | 206.2±37.6 | <0.001 |
| **Triglycerides (mg/dL)** | 95.0 (67.0–141.0) | 77.0 (59.0–104.0) | 102.0 (77.0–134.0) | 159.0 (108.0–209.0) | 176.0 (130.0–238.0) | <0.001 |
| **HDL cholesterol (mg/dL)** | 60.8±16.6 | 67.0±15.8 | 56.6±12.5 | 52.2±14.9 | 47.2±11.7 | <0.001 |
| **LDL cholesterol (mg/dL)** | 126.6±32.6 | 119.6±30.2 | 135.0±30.7 | 134.9±35.1 | 139.5±34.5 | <0.001 |
| **Fasting glucose (mg/dl)** | 94.9±13.8 | 90.9±8.2 | 93.1±7.2 | 104.4±20.2 | 105.5±20.3 | <0.001 |
| **HOMA-IR** | 1.40 (0.95–2.05) | 1.13 (0.80–1.55) | 1.54 (1.15–2.02) | 1.96 (1.35–2.70) | 2.73 (1.98–3.65) | <0.001 |
| **hsCRP (mg/l) (n = 88,275)** | 0.04 (0.03–0.08) | 0.03 (0.02–0.06) | 0.06 (0.04–0.11) | 0.05 (0.03–0.09) | 0.08 (0.05–0.16) | <0.001 |
| **Systolic BP (mmHg)** | 109.5±12.5 | 104.8±10.3 | 113.1±9.9 | 115.3±13.0 | 120.3±12.3 | <0.001 |
| **Diastolic BP (mmHg)** | 71.0±9.8 | 67.7±8.2 | 72.2±8.1 | 76.5±10.4 | 78.8±10.0 | <0.001 |
| **Measured FEV1 (liter)** | 3.32±0.68 | 3.50±0.62 | 3.56±0.66 | 3.20±0.67 | 3.35±0.66 | <0.001 |
| **FEV1%** | 97.7±10.7 | 98.2±10.6 | 98.4±10.7 | 95.6±10.6 | 96.1±10.8 | <0.001 |
| **Measured FVC (liter)** | 4.04±0.86 | 4.31±0.78 | 4.38±0.84 | 3.86±0.85 | 4.15±0.83 | <0.001 |
| **FVC%** | 97.9±10.7 | 98.4±10.7 | 98.9±10.7 | 95.7±10.6 | 96.5±10.7 | <0.001 |
| **FEV1(L)/FVC(L) ratio** | 0.83±0.06 | 0.82±0.05 | 0.81±0.06 | 0.83±0.06 | 0.81±0.05 | <0.001 |
| **Diabetes** | 2,227 (2.0) | 266 (0.4) | 77 (0.4) | 794 (6.6) | 1,090 (6.2) | <0.001 |
| **Hypertension** | 8,241 (7.2) | 1,235 (1.8) | 630 (3.7) | 2,302 (19.2) | 4,074 (23.0) | <0.001 |
| **Dyslipidemia** | 13,706 (12.0) | 3,693 (5.5) | 1,411 (8.2) | 3,583 (29.8) | 5,019 (28.4) | <0.001 |

Data are presented as mean ± standard deviation, median (interquartile range), or the number of subjects with percentage in parenthesis.

We recorded numbers of subjects with available clinical parameters. Unless otherwise indicated, the available subject number was 114,143.

ALT = alanine aminotransferase; BMI = body mass index; BP = blood pressure; FEV1% = percent predicted forced expiratory volume in 1s; FVC% = percent predicted forced vital capacity; HDL = high-density lipoprotein; HOMA-IR = homeostasis model assessment of insulin resistance; hs-CRP = high-sensitivity C-reactive protein; LDL = low-density lipoprotein; MHNO = metabolically healthy non-obese; MHO = metabolically healthy obese; MUHNO = metabolically unhealthy non-obese; MUHO = metabolically unhealthy obese.

## Odd ratios (ORs) for impairment of lung function

Table 4 presents the results of multivariate logistic regression analysis to investigate the effects of metabolic health and obesity on lung function impairment. According to the fully adjusted

**Table 2.** Adjusted mean values of lung function parameters in the study group classified by metabolic health and obesity status.

| | Category | | | | p value by ANCOVA | Adjusted p value[a] | | | | | |
|---|---|---|---|---|---|---|---|---|---|---|---|
| | MHNO (N = 67,261) (58.9%) | MHO (N = 17,192) (15.1%) | MUHNO (N = 12,005) (10.5%) | MUHO (N = 17,685) (15.5%) | | MHNO vs MHO | MHNO Vs MUHNO | MHNO vs MUHO | MHO vs MUHNO | MHO Vs MUHO | MUHNO vs MUHO |
| FEV1 (liter) | 3.306±0.005 | 3.332±0.004 | 3.217±0.005 | 3.289±0.005 | <0.001 | <0.001 | <0.001 | 0.011 | <0.001 | <0.001 | <0.001 |
| FEV1% | 97.822±0.140 | 98.353±0.106 | 95.779±0.150 | 97.281±0.144 | <0.001 | 0.001 | <0.001 | 0.005 | <0.001 | <0.001 | <0.001 |
| FVC (liter) | 4.049±0.006 | 4.074±0.004 | 3.891±0.006 | 4.031±0.006 | <0.001 | <0.001 | <0.001 | 0.034 | <0.001 | <0.001 | <0.001 |
| FVC% | 98.883±0.142 | 99.311±0.104 | 95.593±0.148 | 98.354±0.138 | <0.001 | 0.004 | <0.001 | 0.005 | <0.001 | <0.001 | <0.001 |
| FEV1 (L)/FVC (L) ratio | 0.824±0.001 | 0.815±0.001 | 0.831±0.001 | 0.820±0.001 | <0.001 | <0.001 | <0.001 | <0.001 | <0.001 | <0.001 | <0.001 |

Data are presented as adjusted mean ± standard error. The multivariable model was adjusted for age, sex, and continuous variables with p <0.05 in univariate analyses, comprising systolic blood pressure, smoking (pack-years), alcohol consumption (g/day), moderate physical activity frequency (times/week), glucose, lipid profiles, liver enzymes, creatinine, high-sensitivity C-reactive protein, and HOMA-IR.

[a]Adjusted p value using Bonferroni correction.

FEV1% = percent predicted forced expiratory volume in 1s; FVC% = percent predicted forced vital capacity; HOMA-IR = homeostasis model assessment of insulin resistance; MHNO = metabolically healthy non-obese; MHO = metabolically healthy obese; MUHNO = metabolically unhealthy non-obese; MUHO = metabolically unhealthy obese.

logistic regression analysis, MUH was associated with decreased FEV1% and FVC% (aOR = 1.295 [1.151–1.457] and 1.392 [1.226–1.580], respectively). However, results for obesity showed the opposite pattern. Consequently, the risk of impairment of FEV1% and FVC% was lower in the MHO group compared with MHNO as the reference group. Specifically, aORs (95% CI) for impairment of FEV1% in MHO, MUHNO, and MUHO were 0.871 (0.775–0.978), 1.274 (1.114–1.456), and 1.176 (1.012–1.366), respectively (p for trend = 0.014). Similar results were observed for impairment of FVC% (p for trend = 0.013). The aORs for OLF in MHO, MUHNO, and MUHO compared with MHNO (reference group) were 1.190 (0.959–1.477), 0.904 (0.728–1.123), and 1.275 (1.055–1.539), respectively. However, the difference in aORs for obstructive patterns between groups was not statistically significant (p for trend = 0.173).

## Discussion

In this study, we found that (1) MHO can attenuate impairment of lung function; (2) MUH was significantly associated with impaired lung function, while the reverse pattern was evident for obesity; and (3) obesity was a predictor of OLF but, not MUH.

The prevalence of MHO greatly varies (10–51%) according to race and definitions [1]. MHO is more common in women, younger individuals, and Asian populations compared with Europeans or individuals of multiethnic origin [1]. The prevalence of MHO (15.1%) assessed using criteria proposed by Wildman et al. [20] was comparable to that (11.3–18.2%) reported in the Korean general population using the same criteria [18, 19], whereas this proportion was lower than in a previous study (25.8%) using a less strict definition [24]. Additionally, other studies found lower estimates (7.9%) using a very strict definition [17]. Even in samples of the same ethnicity, MHO prevalence varies depending on the definition. However, there is no consensus on the definition of MHO. Urgent establishment of a common definition of MHO is necessary to evaluate robustly its impact on clinical outcomes.

**Table 3. Comparisons of demographic and clinical parameters according to metabolic health and obesity status.**

|  | All Subjects (N = 114,143) | MHNO (N = 67,261) | MHO (N = 17,192) | MUHNO (N = 12,005) | MUHO (N = 17,685) | p value |
|---|---|---|---|---|---|---|
| **Age (≥40 years)** | 49,240 (43.1) | 25,577 (38.0) | 6,927 (40.3) | 7,364 (61.3) | 9,372 (53.0) | <0.001 |
| **Sex (male)** | 65,342 (57.2) | 28,653 (42.6) | 13,226 (76.9) | 8,671 (72.2) | 14,792 (83.6) | <0.001 |
| **BMI (≥25 kg/m$^2$)** | 34,877 (30.6) | 0 (0.0) | 17,192 (100.0) | 0 (0.0) | 17,685 (100.0) | <0.001 |
| **Metabolic derangement** | 29,690 (26.0) | 0 (0.0)) | 0 (0.0) | 12,005 (100.0) | 17,685 (100.0) | <0.001 |
| **Current smokers** | 18,991 (16.6) | 7,963 (11.8) | 3,482 (20.3) | 2,673 (22.3) | 4,873 (27.6) | <0.001 |
| **Heavy alcohol intake (>20 g/day)** | 22,746 (19.9) | 9,712 (14.4) | 4,213 (24.5) | 3,149 (26.2) | 5,672 (32.1) | <0.001 |
| **Regular exercise (≥3 times/week)** | 15,346 (13.5) | 8,833 (13.2) | 2,864 (16.7) | 1,460 (12.2) | 2,189 (12.4) | <0.001 |
| **High education (≥college education)** | 94,181 (84.3) | 56,457 (85.5) | 14,339 (85.3) | 9,394 (80.2) | 13,991 (81.2) | <0.001 |
| **Elevated bilirubin (>1.9 mg/dL)** | 1,564 (1.4) | 948 (1.4) | 256 (1.5) | 152 (1.3) | 208 (1.2) | 0.038 |
| **Elevated ALT (> 40 U/L) (n = 113,969)** | 11,270 (9.9) | 2,200 (3.3) | 2,298 (13.4) | 1,397 (11.6) | 5,375 (30.4) | <0.001 |
| **Elevated serum creatinine (>1.2 mg/dL)** | 454 (0.4) | 136 (0.2) | 82 (0.5) | 76 (0.6) | 160 (0.9) | <0.001 |
| **Hypercholesterolemia (≥220 mg/dL)** | 23,491 (20.6) | 10,278 (15.3) | 3,991 (23.2) | 3,483 (29.0) | 5,739 (32.5) | <0.001 |
| **Hypertriglyceridemia (≥150 mg/dL)** | 25,290 (22.2) | 4,052 (6.0) | 2,578 (15.0) | 6,870 (57.2) | 11,790 (66.7) | <0.001 |
| **Low HDL cholesterol[a]** | 12,759 (11.2) | 2,117 (3.1) | 800 (4.7) | 3,721 (31.0) | 6,121 (34.6) | <0.001 |
| **High LDL cholesterol (≥ 159 mg/dL)** | 17,001 (14.9) | 6,383 (9.5) | 3,343 (19.4) | 2,697 (22.5) | 4,578 (25.9) | <0.001 |
| **Hyperglycemia at fasting (≥100 mg/dl)** | 25,198 (22.1) | 5,442 (8.1) | 1,772 (10.3) | 7,356 (61.3) | 10,628 (60.1) | <0.001 |
| **HOMA-IR≥90th percentile[b]** | 11,485 (10.1) | 635 (0.9) | 797 (4.6) | 2,256 (18.8) | 7,794 (44.1) | <0.001 |
| **Elevated hsCRP (>0.5 mg/l) (n = 88,275)** | 2,246 (2.5) | 909 (1.8) | 430 (3.1) | 226 (2.5) | 681 (5.0) | <0.001 |
| **High SBP** | 12,981 (11.4) | 2,206 (3.3) | 1,228 (7.1) | 3,461 (28.8) | 6,086 (34.4) | <0.001 |
| **FVC% <80%** | 4,247 (3.7) | 2,183 (3.2) | 462 (2.7) | 686 (5.7) | 916 (5.2) | <0.001 |
| **FEV1% <80%** | 4,878 (4.3) | 2,457 (3.7) | 639 (3.7) | 750 (6.2) | 1,032 (5.8) | <0.001 |
| **FEV1(L)/FVC(L) ratio <0.7** | 2,121 (1.9) | 1,009 (1.5) | 477 (2.8) | 222 (1.8) | 413 (2.3) | <0.001 |

Data are presented as mean ± standard deviation, median and (interquartile range), or the number of subjects with percentage in parentheses.

We recorded subject numbers with available clinical parameters. Unless otherwise indicated, the available subject number was 114,143.

[a]Low HDL was defined as <40 mg/dL in male and <50 mg/d/L in females.

[b]The value of HOMA-IR ≥ 90th percentile is 2.9.

ALT = alanine aminotransferase; BMI = body mass index; SBP = systolic blood pressure; FEV1% = percent predicted forced expiratory volume in 1s; FVC% = percent predicted forced vital capacity; HDL = high-density lipoprotein; HOMA-IR = homeostasis model assessment of insulin resistance; hs-CRP = high-sensitivity C-reactive protein; LDL = low-density lipoprotein; MHNO = metabolically healthy non-obese; MHO = metabolically healthy obese; MUHNO = metabolically unhealthy non-obese; MUHO = metabolically unhealthy obese.

Although our findings are similar to the results of a previous study [12], there were differences. Compared to our current study, the previous study included many more subjects with overt diabetes and hypertension (more than 30% of the sample), which are associated with impaired lung function [14]. Especially, prevalence of diabetes was 13 times higher in the MUHO group than our current study (11.7% vs 0.9%). Given that hypertension is common among patients with diabetes and the combination of hypertension and diabetes had the strongest negative effect on lung function [14], the low spirometric values in the MUHO group can be explained despite the beneficial role of obesity. In addition, diabetes and hypertension contribute to development of cardiovascular diseases [25]. Consequently, it is possible that previous studies contained more subjects with early cardio-pulmonary disabilities than ours, which could affect the results. Additionally, the previous study adjusted for only a few variables, without adjusting for relevant confounders including systemic inflammation [26] and insulin

**Table 4. Multiple logistic regression analysis of impaired spirometric parameters according to metabolic health and obesity status.**

| | Model 1 | | | Model 2 | | | Model 3 | | |
|---|---|---|---|---|---|---|---|---|---|
| | OR (95% CI) | *p* value | *p* for trend | OR (95% CI) | *p* value | *p* for trend | OR (95% CI) | *p* value | *p* for trend |
| FEV1%<80% | | | <0.001 | | | 0.020 | | | 0.014 |
| MHNO (reference) | 1 | | | 1 | | | 1 | | |
| MHO | 0.900 (0.810–0.999) | 0.049 | | 0.893 (0.803–0.992) | 0.035 | | 0.871 (0.775–0.978) | 0.019 | |
| MUHNO | 1.420 (1.283–1.571) | <0.001 | | 1.225 (1.085–1.383) | 0.001 | | 1.274 (1.114–1.456) | <0.001 | |
| MUHO | 1.406 (1.281–1.543) | <0.001 | | 1.154 (1.007–1.322) | 0.039 | | 1.176 (1.012–1.366) | 0.034 | |
| Metabolic unhealthy | 1.468 (1.371–1.573) | | <0.001 | 1.244 (1.118–1.385) | | <0.001 | 1.295 (1.151–1.457) | | <0.001 |
| Obesity | 0.847 (0.793–0.903) | | <0.001 | 0.904 (0.828–0.986) | | 0.022 | 0.902 (0.820–0.993) | | 0.035 |
| FVC%<80% | | | <0.001 | | | 0.003 | | | 0.013 |
| MHNO (reference) | 1 | | | 1 | | | 1 | | |
| MHO | 0.744 (0.660–0.839) | | <0.001 | 0.739 (0.655–0.833) | <0.001 | | 0.704 (0.615–0.805) | <0.001 | |
| MUHNO | 1.499 (1.346–1.670) | | <0.001 | 1.251 (1.100–1.423) | 0.001 | | 1.241 (1.075–1.432) | 0.003 | |
| MUHO | 1.365 (1.236–1.507) | | <0.001 | 1.248 (1.080–1.442) | 0.003 | | 1.226 (1.043–1.441) | 0.014 | |
| Metabolic unhealthy | 1.635 (1.524–1.753) | | <0.001 | 1.384 (1.235–1.550) | | <0.001 | 1.392 (1.226–1.580) | | <0.001 |
| Obesity | 0.863 (0.804–0.926) | | <0.001 | 0.862 (0.785–0.947) | | 0.002 | 0.835 (0.752–0.928) | | 0.001 |
| FEV1(L)/FVC(L) ratio<0.7 | | | 0.074 | | | 0.062 | | | 0.173 |
| MHNO (reference) | 1 | | | 1 | | | 1 | | |
| MHO | 1.321 (1.135–1.538) | <0.001 | | 1.237(1.015–1.509) | 0.035 | | 1.190 (0.959–1.477) | 0.114 | |
| MUHNO | 0.961 (0.822–1.123) | 0.615 | | 0.905 (0.741–1.105) | 0.327 | | 0.904 (0.728–1.123) | 0.362 | |
| MUHO | 1.395 (1.176–1.655) | <0.001 | | 1.308(1.099–1.556) | 0.002 | | 1.275(1.055–1.539) | 0.012 | |
| Metabolic unhealthy | 1.004 (0.908–1.111) | | 0.932 | 1.055 (0.904–1.230) | | 0.498 | 1.067 (0.902–1.262) | | 0.451 |
| Obesity | 1.371 (1.212–1.550) | | <0.001 | 1.139(1.181–1.517) | | <0.001 | 1.295(1.131–1.484) | | <0.001 |

Model 1 was adjusted for age, sex, smoking status, alcohol intake, regular exercise and education level. Model 2 was adjusted as in model 1 plus systolic BP, glucose, total cholesterol, triglycerides, HDL-C, LDL-C, and HOMA-IR. Model 3 was adjusted as in model 2 plus variables with *p* <0.05 in univariate analyses.

CI = confidence interval; FEV1% = percent predicted forced expiratory volume in 1s; FVC% = percent predicted forced vital capacity; MHNO = metabolically healthy non-obese; MHO = metabolically healthy obese; MUHNO = metabolically unhealthy non-obese; MUHO = metabolically unhealthy obese; OR = odds ratio.

resistance [10], affecting both MUH and lung function impairment. Failing to adjust for those confounders could also distort the results. Finally, we also used new reference equations that are more suitable and more reliable for predicting lung function in Korean samples [22]. Thus, our results might be more relevant to the general population than those of the previous study.

Furthermore, the same association between MHO and lung function was identified using a strict definition of MH as absence of any component of metabolic syndrome [17]. The aOR was attenuated in MUHNO and MUHO in contrast to a stronger aOR value of MHO compared to the original results (S1 Table). The reasons for these results are unclear. However, individuals with MH phenotypes could be more homogeneous, and vice versa when using a strict definition of MH. These changes seemed to impact the effect of MH on lung function more strictly despite an attenuated effect of MUH on lung function. Consequently, a strict definition of MH might have allowed us to identify effects of metabolic health status on lung function that were missed when using the definition of MH as having fewer than 2 metabolic components, because each metabolic parameters is a risk factor of decreased lung function [10, 27]. Therefore, the aOR values might be stronger in the MHO phenotype and weaker in MUHNO and MUHO. These changes would provide more relevant results, as in a previous study assessing the effect of MHO on coronary calcium [17]. Such findings are important to support the benefit of MHO for lung function, because the marked heterogeneity of MHO definitions could cause differences in the association with lung function.

Our study demonstrates the benefit of MHO for lung function. Although the reasons for this result remain unclear, we offer some possible explanations. First, the obesity paradox could explain our finding. Metabolic abnormalities are well known risk factors for impaired lung function [10, 11]. However, the role of obesity in impaired lung function is controversial. We showed that obesity was associated with attenuated decline of lung function, as in previous studies [8, 9, 12], although there is conflicting evidences on the effect of obesity on lung function [5–7]. This discrepancy was caused by differences in the prevalence of BMI $\geq$25 kg/m$^2$ and BMI value. The prevalence of BMI $\geq$25 kg/m$^2$ and median or mean BMI value in studies [8, 9, 12] consistent with ours were lower (12–31% and 21–23.6) than those (57–70% and 26.4–28.2) in studies with conflicting results [5–7]. Therefore, subjects in the former studies were less obese than those in the latter. Interestingly, the obesity paradox most likely applies to moderately obese subjects, especially those with BMI<35kg/m$^2$ [3]. Specifically, the majority (98.6%) of our obese subjects (99.6% in the MHO group) had values of BMI<35kg/m$^2$. Collectively, the obese individuals in this and previous studies [8, 9, 12] were moderately obese, especially in MHO individuals [28] with strong obesity paradox benefits, which could explain our findings. Second, demographic data such as physical activity (PA) can explain our findings. The majority of our subjects was middle aged city dwellers of higher socioeconomic level who invested more interest in their health, especially obese subjects who were aware of the harmful effects of obesity. Accordingly, our MHO subjects exercised the most among the four groups, as in previous studies [17, 18]. The main contributor to metabolic health in the obese population is cardiorespiratory fitness (CRF) [2, 3], which includes critical contributions of muscular fitness, stamina, and strength [2]. Maintaining or taking up PA, which likely results in improved CRF, contributes to muscle augmentation including respiratory muscle [29], which is in turn related to better lung function [30]. In addition, high levels of CRF largely neutralize the adverse effects of obesity. This indicates that CRF is a critical component for understanding the obesity paradox [2, 3]. Taken together, the strong paradoxical benefit related to moderate obesity and increased PA in our MHO subjects might explain the benefits of MHO on lung function.

We also found that FEV1/FVC was notably decreased in the obese population, regardless of metabolic health. Obese individuals might exhibit airway narrowing due to the mechanical effects of obesity and increased pro-inflammatory cytokines linked to airway inflammation [4]. However, the effect of obesity on FEV1/FVC is controversial, with some studies [9, 27, 31] finding adverse effects as in the current study, and others finding the opposite [5, 7, 32, 33]. Although the reasons for these mixed results are unclear, severity of obesity might explain the disagreement. The current study and previous studies [9, 27, 31] were undertaken in moderately obese cohorts with lower median or mean BMI value (23.6–25.1) than in studies showing conflicting results (26.4–28.5) [5, 7, 32, 33]. Abdominal adiposity might restrict the descent of the diaphragm and limit lung expansion [34]. This mechanical impairment triggers reduction of FVC, accompanied by reduction of FEV1 in proportion to FVC [6]. Consequently, FVC decline is more pronounced compared to FEV1, resulting in a positive correlation between obesity and FEV1/FVC [35]. However, moderate obesity is generally believed to have little effect on ventilator lung function [36], and the obesity paradox more likely applies to moderately obese subjects [3]. Therefore, moderate obesity could cause improved FVC rather than impaired FVC. In accordance to changes in FVC associated with moderate obesity, FEV1/FVC might be decreased in our cohort and previous studies [9, 27, 31], leading to obesity as a predictor of OLF. In contrast, MUH was not associated with OLF in the current study, as previously reported [10]. We also found no significant differences between groups divided by metabolic health and obesity status and OLF. A possible explanation for this disagreement is the smaller contribution of MUH to OLF than of obesity in our cohort, because of low levels of

systemic inflammation [26] and insulin resistance [10], which link metabolic derangement and lung function impairment. Specifically, the median values of CRP and HOMA-IR in our MUH group were 0.07 and 2.47, respectively, which were close to the upper normal limits to define systemic inflammation and insulin resistance [16]. Furthermore, functional debility of the airways might have gone undetected on screening spirometry for our healthy subjects because OLF predominantly reflects obstruction of large airways [37]. Therefore, these seem to attenuate the relationship between metabolically healthy and unhealthy obese- or non-obese subjects and OLF, in contrast to the relationships of OLF in obese- and non-obese subjects.

In the current study, we demonstrated that MHO does not have harmful effects on lung function, explained by the obesity paradox. Reduced lung function is a major risk factor for COPD and cardiovascular morbidity and mortality, which are potentially preventable diseases with significant health and economic impacts worldwide [38]. Furthermore, reduced lung function is also a powerful predictor of mortality [39]. Therefore, the current study emphasizes the importance of demographic indicators, the obesity paradox, and healthy weight for lung function, given the projected growing public health impact of lung function [38, 39] and high prevalence of obesity [1, 2].

A major strength of our study is its large sample size and adjustments for multiple confounding factors related to lung function. This gave us sufficient statistical power to determine the relationship between MHO and lung function. However, our study has several limitations. First, the cross-sectional design precludes the determination of causality. Hence, further studies are needed to validate our findings. However, we studied asymptomatic young individuals from a large health screening cohort, minimizing the possibility of reverse causation. Second, our results were obtained in a middle aged Korean health screening cohort with moderate obesity, which means it is not possible to generalize our findings. The final limitation is the inability of BMI to differentiate between fat and muscle, which have opposite effects on lung function [4]. As a parameter of obesity, BMI frequently categorized people as obese even if they have higher muscle-to-fat-tissue ratios, especially in asymptomatic middle-aged healthy subjects like those in our cohort [40]. We could not rule out the effects of confounding factors.

## Conclusions

We found that MHO was associated with attenuated FVC and FEV1 decline in a middle-aged Korean sample. The association seems to be related to the obesity paradox, the effect of which is particularly strong in our cohort. These findings suggest maintaining healthy weight and life style could be more important to improve lung function than obesity defined by only BMI. However, longitudinal follow-up studies and prospective interventional studies should be required to validate any cause-and-effect relationships between MHO and lung function.

## Supporting information

**S1 Table. Multiple logistic regression analysis of impaired spirometric parameters according to metabolic health defined as absence of any component of metabolic syndrome and obesity status.**
(DOCX)

## Author Contributions

**Conceptualization:** Jonghoo Lee, Jae-Uk Song.

**Data curation:** Soo-Youn Ham, Si-Young Lim, Jae-Uk Song.

**Formal analysis:** Jonghoo Lee, Hye Kyeong Park, Jae-Uk Song.

**Funding acquisition:** Jonghoo Lee.

**Investigation:** Jae-Uk Song.

**Methodology:** Min-Jung Kwon, Jae-Uk Song.

**Project administration:** Jae-Uk Song.

**Resources:** Jae-Uk Song.

**Software:** Jae-Uk Song.

**Supervision:** Jae-Uk Song.

**Validation:** Jae-Uk Song.

**Visualization:** Jae-Uk Song.

**Writing – original draft:** Jonghoo Lee, Hye Kyeong Park, Jae-Uk Song.

**Writing – review & editing:** Jonghoo Lee, Hye Kyeong Park, Min-Jung Kwon, Soo-Youn Ham, Si-Young Lim, Jae-Uk Song.

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
