## [Decision Letter · Decision Letter 0]

2 Dec 2021

PONE-D-21-23871

Metabolically-healthy obesity is not detrimental to lung function: A Kangbuk Samsung Health Study

PLOS ONE

Dear Dr. Song,

Thank you for submitting your manuscript to PLOS ONE. After careful consideration, we feel that it has merit but does not fully meet PLOS ONE’s publication criteria as it currently stands. Therefore, we invite you to submit a revised version of the manuscript that addresses the points raised during the review process.

Please address carefully the reviewer comments. Please ensure that the conclusions are supported by the data. In the current stage the manuscript is misleading and requires significant changes.

We look forward to receiving your revised manuscript.

Kind regards,

Anna Halama

Academic Editor

PLOS ONE

“This work was supported by 2022 scientific promotion program funded by Jeju National University”

“One of authors(Jonghoo Lee) was supported by 2022 scientific promotion program funded by Jeju National University”

Reviewers' comments:

Reviewer's Responses to Questions

**Comments to the Author**

1. Is the manuscript technically sound, and do the data support the conclusions?

Reviewer #1: Partly

2. Has the statistical analysis been performed appropriately and rigorously? 

Reviewer #1: Yes

3. Have the authors made all data underlying the findings in their manuscript fully available?

Reviewer #1: Yes

4. Is the manuscript presented in an intelligible fashion and written in standard English?

Reviewer #1: Yes

5. Review Comments to the Author

Reviewer #1: Thank you for allowing me to review your paper.

Song and colleagues address whether metabolically healthy obesity influences lung functions based on a cohort of 114,143 people recruited by a cross-sectional study from a medical health checkup program in South Korea. The participants were divided into four groups: MHNO (as reference), MHO, MUHNO and MUHO. Results suggested that MHO group had better lung functions compared with other groups. The study was generally well conducted. Comments are provided as follow:

Major comment

The title and conclusion are not supported by the study design. The authors suggested that MHO group is not always harmful, but this is solely based on cross-sectional data. It is likely reverse causation may play a role here. The discussion suggested the use of asymptomatic young individuals minimized the possibility of reverse causation which is not completely convincing. E.g., people with subclinical lung disease may also lose weight because of those diseases. Therefore, I strongly recommend the title and conclusion to be revised to be more neutral.

Minor comments

1. The OR values should be included in the Abstract.

2. Sensitivity analysis of stricter definition criteria of metabolic health can be considered to see the findings are still consistent. For instance, initially meet two metabolic syndrome components as MHO, now meet four.

3. The reporting of the study design can be better organized. Covariates, exposures, and outcomes should be described under the corresponding headings. For example, blood samples should be described under metabolic status, and covariates should be written with subheadings.

4. Figure 2 can be presented in a table instead of a graph.

5. The definition of metabolic derangement should be provided.

6. PLOS authors have the option to publish the peer review history of their article (what does this mean?). If published, this will include your full peer review and any attached files.

Reviewer #1: No

---

## [Author Response · Author response to Decision Letter 0]

11 Jan 2022

The authors appreciate the time and effort given by all of the reviewers and the editor in reviewing this manuscript. The constructive criticism has been well-received by all of us. Thank you for the constructive comments and criticisms of our manuscript named above. In response to the reviewer’s suggestions and questions, we have revised the manuscript as suggested. We have responded to the reviewer’s comments in a point-by-point manner below. Please find enclosed the revised version of the manuscript. We hope we have dealt with comments of the reviewers to their satisfaction. We hope that this work is now suitable for publication and look forward to your response.

PS) One of authors (Jonghoo Lee) was supported by 2022 scientific promotion program funded by Jeju National University.However, the funders had no role in study design, data collection and analysis, decision to publish, or preparation of the manuscript.

Your assistance will be greatly appreciated.

With best wishes

---

## [Decision Letter · Decision Letter 1]

16 Feb 2022

PONE-D-21-23871R1Manuscript ID: PONE-D-21-23871 Title: The effect of metabolically-health and obesity on lung function: A cross sectional study of 114,143 participants from Kangbuk Samsung Health StudyPLOS ONE

Dear Dr. Song,

Thank you for submitting your manuscript to PLOS ONE. After careful consideration, we feel that it has merit but does not fully meet PLOS ONE’s publication criteria as it currently stands. Therefore, we invite you to submit a revised version of the manuscript that addresses the points raised during the review process. Please address the reviewers comments. Please ensure to conduct English proofreading.

We look forward to receiving your revised manuscript.

Kind regards,

Anna Halama

Academic Editor

PLOS ONE

Journal Requirements:

Reviewers' comments:

Reviewer's Responses to Questions

**Comments to the Author**

1. If the authors have adequately addressed your comments raised in a previous round of review and you feel that this manuscript is now acceptable for publication, you may indicate that here to bypass the “Comments to the Author” section, enter your conflict of interest statement in the “Confidential to Editor” section, and submit your "Accept" recommendation.

Reviewer #1: All comments have been addressed

2. Is the manuscript technically sound, and do the data support the conclusions?

Reviewer #1: Yes

3. Has the statistical analysis been performed appropriately and rigorously? 

Reviewer #1: Yes

4. Have the authors made all data underlying the findings in their manuscript fully available?

Reviewer #1: Yes

5. Is the manuscript presented in an intelligible fashion and written in standard English?

Reviewer #1: Yes

6. Review Comments to the Author

Reviewer #1: Thanks for the authors’ revision.

The author has addressed all the points I mentioned before and interpreted them reasonably and comprehensively.

There are some minor comments attached.

1. It would be better to do some English proofreading to make the manuscript easier to read.

2. As for R2, the paragraph simply explains why a stricter definition would give the most conservative result. It would be transparent to show the readers if the OR would still be relevant or even stronger.

7. PLOS authors have the option to publish the peer review history of their article (what does this mean?). If published, this will include your full peer review and any attached files.

Reviewer #1: No

---

## [Author Response · Author response to Decision Letter 1]

25 Feb 2022

The authors appreciate the time and effort of the reviewers and the editor in reviewing this manuscript and in providing thoughtful and helpful comments. We are pleased to have an opportunity to improve our paper. The constructive criticism has been well-received by all of us. In response to the reviewer’s suggestions and questions, we have revised the manuscript as suggested. We have responded to the reviewer’s comments in a point-by-point manner below. We have included the changes recommended by the editors and reviewers in the revised manuscript (in red). Please find enclosed the revised version of the manuscript. We believe that we have dealt with all reviewer comments and hope that this work is now suitable for publication. 

We look forward to your response.

With best wishes

---

## [Editor Report · Decision Letter 2]

30 Mar 2022

Manuscript ID: PONE-D-21-23871 Title: The effect of metabolic health and obesity on lung function: A cross sectional study of 114,143 participants from Kangbuk Samsung Health Study

PONE-D-21-23871R2

Dear Dr. Song,

We’re pleased to inform you that your manuscript has been judged scientifically suitable for publication and will be formally accepted for publication once it meets all outstanding technical requirements.

Kind regards,

Anna Halama

Academic Editor

PLOS ONE
---

## [Editor Report · Acceptance letter]

4 Apr 2022

PONE-D-21-23871R2 

The effect of metabolic health and obesity on lung function: A cross sectional study of 114,143 participants from Kangbuk Samsung Health Study 

Dear Dr. Song:

I'm pleased to inform you that your manuscript has been deemed suitable for publication in PLOS ONE. Congratulations! Your manuscript is now with our production department. 

Kind regards, 

on behalf of

Dr. Anna Halama 

Academic Editor

PLOS ONE